# ROBUST OVERFITTING MAY BE MITIGATED BY PROPERLY LEARNED SMOOTHENING

**Tianlong Chen[1*], Zhenyu Zhang[2*], Sijia Liu[3,4], Shiyu Chang[4], Zhangyang Wang[1]**
[1]University of Texas at Austin,[2]University of Science and Technology of China
[3]Michigan State University, [4]MIT-IBM Watson AI Lab, IBM Research
{tianlong.chen,atlaswang}@utexas.edu, zzy19969@mail.ustc.edu.cn
liusiji5@msu.edu, shiyu.chang@ibm.com

## ABSTRACT

A recent study (Rice et al., 2020) revealed overfitting to be a dominant phenomenon in adversarially robust training of deep networks, and that appropriate early-stopping of adversarial training (AT) could match the performance gains of most recent algorithmic improvements. This intriguing problem of *robust overfitting* motivates us to seek more remedies. As a pilot study, this paper investigates two empirical means to inject more *learned smoothening* during AT: one leveraging knowledge distillation and self-training to *smooth the logits*, the other performing stochastic weight averaging (Izmailov et al., 2018) to *smooth the weights*. Despite the embarrassing simplicity, the two approaches are surprisingly effective and hassle-free in mitigating robust overfitting. Experiments demonstrate that by plugging in them to AT, we can simultaneously boost the standard accuracy by $3.72\% \sim 6.68\%$ and robust accuracy by $0.22\% \sim 2.03\%$, across multiple datasets (STL-10, SVHN, CIFAR-10, CIFAR-100, and Tiny ImageNet), perturbation types ($\ell_\infty$ and $\ell_2$), and robustified methods (PGD, TRADES, and FSGM), establishing the new state-of-the-art bar in AT. We present systematic visualizations and analyses to dive into their possible working mechanisms. We also carefully exclude the possibility of gradient masking by evaluating our models' robustness against transfer attacks. Codes are available at https://github.com/VITA-Group/Alleviate-Robust-Overfitting.

## 1 INTRODUCTION

Adversarial training (AT) (Madry et al., 2018), i.e., training a deep network to minimize the worst-case training loss under input perturbations, is recognized as the current best defense method to adversarial attacks. However, one of its pitfalls was exposed by a recent work (Rice et al., 2020): in contrast to the commonly-held belief that overparameterized deep networks hardly overfit in standard training (Zhang et al., 2016; Neyshabur et al., 2017; Belkin et al., 2019), overfitting turns out to be a dominant phenomenon in adversarially robust training of deep networks. After a certain point in AT, e.g., immediately after the first learning rate decay, the robust test errors will only continue to substantially increase with further training (see Figure 1 bottom for example).

That surprising phenomenon, termed as *"robust overfitting"*, has been prevalent on many datasets and models. As Rice et al. (2020) pointed out, it poses serious challenges to assess recent algorithmic

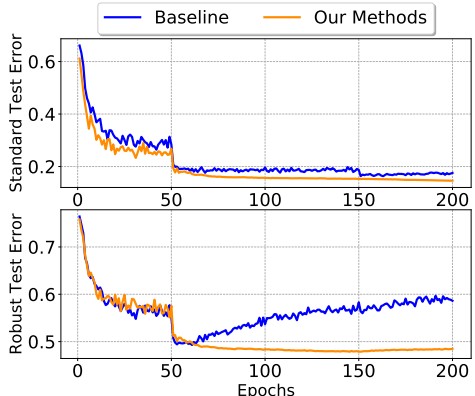

Figure 1: The standard (top) and robust (bottom) test error rate comparison, between the vanilla PGD-AT baseline (Madry et al., 2018), and PGD-AT with our proposed weight/label smoothing techniques applied, on CIFAR-10 with ResNet-18. Our methods effectively mitigates the *robust overfitting* (Rice et al., 2020) even when trained to 200 epochs, while maintaining the same high standard/robust accuracies compared to the *best early-stop checkpoint* of the baseline.

---

*Equal Contribution.

advances upon AT: by just using an earlier checkpoint, the performance of AT be drastically boosted to match the more recently reported state-of-the-arts (Yang et al., 2019b; Zhang et al., 2019b). Even worse, Rice et al. (2020) tested several other implicit and explicit regularization methods, including weight decay, data augmentation and semi-supervised learning; they reported that none of those alternatives seem to combat robust overfitting (stably) better than simple early stopping. The authors thus advocated using the validation set to select a stopping point, although the manual picking would inevitably trade off between selecting either the peak point of robust test accuracy or that of standard accuracy, which often do not coincide (Chen et al., 2020a).

*Does there exist more principled, hands-off, and hassle-free mitigation for this robust overfitting, for us to further unleash the competency of AT*? This paper explores two options along the way, that draw two more sophisticated ideas from enhancing standard deep models' generalization. Both could be viewed as certain types of *learned smoothening*, and are directly plugged into AT:

- Our first approach is to *smooth the logits* in AT via self-training, using knowledge distillation with the same model pre-trained as a self-teacher. The idea is inspired by two facts: (1) label smoothening (Szegedy et al., 2016) can calibrate the notorious overconfidence of deep networks (Hein et al., 2019), and that was found to improve their standard generalization; (2) label smoothening can be viewed as a special case of knowledge distillation (Yuan et al., 2020), and self-training can produce more semantic-aware and discriminative soft label "self-teachers" than naive label smoothening (Chen et al., 2020b; Tang et al., 2020).

- Our second approach is to *smooth the weights* in AT via stochastic weight averaging (SWA) (Izmailov et al., 2018), a popular training technique that leads to better standard generalization than SGD, with almost no computational overhead. While SWA has not yet be applied to AT, it is known to find flatter minima which are widely believed to indicate stronger robustness (Hein & Andriushchenko, 2017; Wu et al., 2020a). Meanwhile, SWA could also be interpreted as a temporal model ensemble, and therefore might bring the extra robustness of ensemble defense (Tramèr et al., 2018; Grefenstette et al., 2018) with the convenience of a single model. Those suggest that applying SWA is natural and promising for AT.

To be clear, neither knowledge-distillation/self-training nor SWA was invented by this paper: they have been utilized in standard training to alleviate (standard) overfitting and improve generalization, by fixing over-confidence and by finding flatter solutions, respectively. By introducing and adapting them to AT, **our aim** is to complement the existing study, demonstrating that while simpler regularizations were unable to fix robustness overfitting as Rice et al. (2020) found, our learned logit/weight smoothening could effectively regularize and mitigate it, without needing early stopping.

Experiments demonstrate that by plugging in the two techniques to AT, we can simultaneously boost the standard accuracy by $3.72\% \sim 6.68\%$ and robust accuracy by $0.22\% \sim 2.03\%$, across multiple datasets (STL-10, SVHN, CIFAR-10, CIFAR-100, and Tiny ImageNet), perturbation types ($\ell_\infty$ and $\ell_2$), and robustified methods (PGD, TRADES, and FSGM), establishing the new state-of-the-art in AT. As shown in Figure 1 example, our method eliminates the robust overfitting phenomenon in AT, even when training up to 200 epochs. Our results imply that although robustness overfitting is more challenging than standard overfitting, its mitigation is still feasible with properly-chosen, advanced regularizations that were developed for the latter. Overall, our findings join (Rice et al., 2020) in re-establishing the competitiveness of the simplest AT baseline.

## 1.1 BACKGROUND WORK

Deep networks are easily fooled by imperceivable adversarial samples. To tackle this vulnerability, numerous defense methods were proposed (Goodfellow et al., 2015; Kurakin et al., 2016; Madry et al., 2018), yet many of them (Liao et al., 2018; Guo et al., 2018; Xu et al., 2017; Dziugaite et al., 2016; Dhillon et al., 2018; Xie et al., 2018; Jiang et al., 2020) were later found to result from training artifacts, such as obfuscated gradients (Athalye et al., 2018) caused by input transformation or randomization. Among them, adversarial training (AT) (Madry et al., 2018) remains one of the most competitive options. Recently more improved defenses have been reported (Dong et al., 2018; Yang et al., 2019b; Mosbach et al., 2018; Hu et al., 2020; Wang et al., 2020a; Dong et al., 2020; Zhang et al., 2020a;b), with some of them also being variants of AT, e.g. TRADES (Zhang et al., 2019b) and AT with metric learning regularizers (Mao et al., 2019; Pang et al., 2019; 2020).

While overfitting has become less a practical concern in training deep networks nowadays, it was not yet noticed nor addressed in the adversarial defense field until lately. An overfitting phenomenon was

first observed in a few fast adversarial training methods (Zhang et al., 2019a; Shafahi et al., 2019b; Wong et al., 2020) based on FGSM (Goodfellow et al., 2015), e.g., sometimes the robust accuracy against a PGD adversary suddenly drop to nearly zero after some training. (Andriushchenko & Flammarion, 2020) suggested it to be rooted in those methods' local linearization assumptions of the loss landscape in those "fast" AT. The recently reported *robust overfitting* (Rice et al., 2020) seems to raise a completely new challenge for the classical AT (not fast): the model starts to irreversibly lose robustness after training with AT for a period, even the double-descent generalization curves still seemed to hold (Belkin et al., 2019; Nakkiran et al., 2019). Among various options tried in Rice et al. (2020), early-stopping was so far the only effective remedy found.

## 2 METHODOLOGY

### 2.1 LEARNING TO SMOOTH LOGITS IN AT

**Rationale: Why**  AT enforces models robust against adversarial attacks of a specific type and certain magnitudes. However, it has been shown to "overfit" the threat model "seen" during training (Kang et al., 2019; Maini et al., 2019; Stutz et al., 2020), and its gained robustness does not extrapolate to larger perturbations nor unseen attack types. Stutz et al. (2020) hypothesized this to be an unwanted consequence of enforcing high-confidence predictions on adversarial examples since high-confidence predictions are difficult to extrapolate to arbitrary regions beyond the seen examples during training. We generalize this observation: during AT, the attacks generated at every iteration can be naturally considered as continuously varying/evolving, along with the model training. Therefore, we hypothesize one source of robust overfitting might lie in that the model "overfits" the attacks generated in the early stage of AT and fails to generalize or adapt to the attacks in the late stage.

To alleviate the overconfidence problem, we adapt the label smoothening (LS) technique in standard training (Szegedy et al., 2016). LS creates uncertainty in the one-hot labels, by computing cross-entropy not with the "hard" targets from the dataset, but with a weighted mixture of these one-hot targets with the uniform distribution. This uncertainty helps to tackle alleviate the overconfidence problem Hein et al. (2019) and improves the standard generalization. The idea of LS was previously investigated in other defense methods (Shafahi et al., 2019a; Goibert & Dohmatob, 2019), but much of the observed robustness gains were later attributed to obfuscated gradients (Athalye et al., 2018). Two recent works (Stutz et al., 2020; Cheng et al., 2020) have integrated LS with AT to inject label uncertainty: Stutz et al. (2020) used a convex combination of uniform and one-hot distributions as target for the cross-entropy loss in AT, which resembles the LS regularizer, while Cheng et al. (2020) concurrently used an LS regularizer for AT.

However, there is one pitfall of the naive LS in (Szegedy et al., 2016): over-smoothening labels in a data-blind way could cause loss of information in the logits, and hence weakened discriminative power of the trained models (Müller et al., 2019). That calls for a careful and adaptive balance between discriminative capability and confidence calibration of the model. In the context of AT, Stutz et al. (2020) crafted a perturbation-dependent parameter, to explicitly control the transition from one-hot to the uniform distribution when the attack magnitude grows from small to large. To identify more automated and principled means, we notice another recent work (Yuan et al., 2020), who explicitly connected knowledge distillation (KD) (Hinton et al., 2015) to LS. The authors pointed out that LS equals a special case of KD using a virtual and hand-crafted teacher; on the contrary, the conventional KD provides data-driven soften labels rather than simply mixing one-shot and uniform vectors. Together with many others (Furlanello et al., 2018; Chen et al., 2020b), these works demonstrated that using model-based and learned soft labels supplies much superior confidence calibration and logit geometry compared to the naive LS (Tang et al., 2020).

Furthermore, (Furlanello et al., 2018; Chen et al., 2020b; Yuan et al., 2020) unanimously revealed that another strong teacher model with extra privileged information is NOT critical to the success of KD. Yuan et al. (2020) shows that even a poorly-trained teacher with much lower accuracy can still improve the student. Moreover, Chen et al. (2020b); Yuan et al. (2020) find *self-teacher* to be sufficiently effective for KD, that is, using soft-logit outputs from the student or designed manually as the KD regularization to train itself (also called teacher-free KD

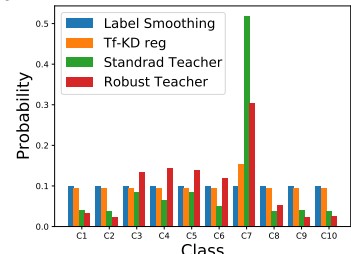

Figure 2: Comparing the logit distribution of different LS/KD means on CIFAR-10 using ResNet-18 (C7 is the correct label).

(Tf-KD) in (Yuan et al., 2020)). These observations make the main cornerstone for our learned logit smoothening approach next.

**Approach: How**    We follow (Chen et al., 2020b; Yuan et al., 2020) to use self-training with the same model, but introduce one specific modification. The one model could be trained with at least two different ways: standard training, or robust training (AT or other cheaper ways; see ablation experiments). That can yield two self-teachers. We assume both to be available; and let $\mathbf{x}$ be the input, $y$ the one-hot ground truth label, $\boldsymbol{\delta}$ the adversarial perturbation bounded by $\ell_p$ norm ball with radius $\epsilon$, and $\boldsymbol{\theta}_r/\boldsymbol{\theta}_s$ the weights of the robust-/standard-trained self-teachers, respectively. Note the two self-teachers share the identical network architecture and training data with our target model. Our self-training smoothed loss function is expressed below ($\lambda_1$ and $\lambda_2$ are two hyperparameters):

$$\min_{\boldsymbol{\theta}} \quad \mathbb{E}_{(\mathbf{x},y)\in\mathcal{D}} \left\{ (1-\lambda_1-\lambda_2) \cdot \max_{\boldsymbol{\delta}\in\mathrm{B}_\epsilon(\mathbf{x})} \mathcal{L}_{\mathrm{XE}}(f(\boldsymbol{\theta},\mathbf{x}+\boldsymbol{\delta}),y) + \right.$$
$$\left. \lambda_1 \cdot \mathcal{KD}_{\mathrm{adv}}(f(\boldsymbol{\theta},\mathbf{x}+\boldsymbol{\delta}),f(\boldsymbol{\theta}_r,\mathbf{x}+\boldsymbol{\delta})) + \lambda_2 \cdot \mathcal{KD}_{\mathrm{std}}(f(\boldsymbol{\theta},\mathbf{x}+\boldsymbol{\delta}),f(\boldsymbol{\theta}_s,\mathbf{x}+\boldsymbol{\delta})) \right\}, \quad (1)$$

where $\mathcal{L}_{\mathrm{XE}}$ is robustified cross-entropy loss adopted in the original AT; $\mathcal{KD}_{\mathrm{adv}}$ and $\mathcal{KD}_{\mathrm{std}}$ are the Kullback–Leibler divergence loss with the robust-trained and standard-trained self-teachers, respectively. $\lambda_1 = 0.5$ and $\lambda_2 = 0.25$ are default in all experiments. More details are in Appendix A2.1.

Figure 2 visualizes an example of logit distributions, generated by naive LS (Szegedy et al., 2016), the Tf-KD regularizer using manually-designed self-teacher in (Yuan et al., 2020), as well our standard- and robust-trained teachers, respectively. We observe both standard and robust self-teachers are more discriminative than the other two baseline smoothenings, while the robust self-teacher is relatively more conservative as one shall expect.

## 2.2    Learning to Smooth Weights in AT

**Rationale: Why**    Another measure that is often believed to indicate the standard generalization is the *flatness*: the loss surface at the final learned weights for well-generalizing models is relatively "flat". Similarly, Wu et al. (2020a) advocated that a flatter adversarial loss landscape shrinks the robustness generalization gap. This is aligned with (Hein & Andriushchenko, 2017) where the authors called it local Lipschitz and proved that the Lipschitz constant can be used to formally measure the robustness of machine learning models. The flatness preference of a robust model has been echoed by many empirical defense methods, such as hessian/curvature-based regularization (Moosavi-Dezfooli et al., 2019), gradient magnitude penalty (Wang & Zhang, 2019), smoothening with random noise (Liu et al., 2018), or entropy regularization (Jagatap et al., 2020). However, all those methods will incur (sometimes heavy) computational or memory overhead; and many can cause standard accuracy drops, e.g., hessian/curvature-based methods (Gupta et al., 2020).

Stochastic weight averaging (SWA) (Izmailov et al., 2018) was proposed to enforce the weight smoothness, by simply averaging multiple checkpoints along the training trajectory. SWA is known to find much flatter solutions than SGD, is extremely easy to implement, improves standard generalization, and has almost no computational overhead. SWA has been successfully adopted in semi-supervised learning (Athiwaratkun et al., 2018), Bayesian inference (Maddox et al., 2019), and low-precision training (Yang et al., 2019a). In this paper, we introduce SWA to AT for the first time, in order to smooth the weights and find flatter minima that may improve the adversarially robust generalization. Note that we choose SWA mainly due to its simplicity for proofs-of-concept; while extensively comparing alternative "flatness" regularizations is beyond our current work's scope.

One additional bonus of adopting SWA in AT is the *temporal ensemble* effect of SWA. It has been widely observed (Tramèr et al., 2018; Grefenstette et al., 2018; Wu et al., 2020b; Wang et al., 2021) that training a model with the attack transferred from another could reduce "trivial robustness" caused by locally nonlinear loss surfaces, and therefore constructed model ensembles for a stronger defense. SWA was interpreted as approximating the fast geometric ensembling (Garipov et al., 2018), by aggregating multiple checkpoint weights at different training time. Applying SWA to AT therefore may lead to stronger and more transferable attacks, and consequently stronger defense due to ensembling, with the convenience of a single model.

**Approach: How**    Following (Izmailov et al., 2018), applying SWA to AT is straightforward:

$$\mathcal{W}_{\mathrm{SWA}}^{\mathrm{T}} = \frac{\mathcal{W}_{\mathrm{SWA}}^{\mathrm{T-1}} \times n + \mathcal{W}^{\mathrm{T}}}{n+1}, \quad \mathcal{W}^{\mathrm{T}} = \mathcal{W}^{\mathrm{T-1}} + \Delta\mathcal{W}^{\mathrm{T}} \quad (2)$$

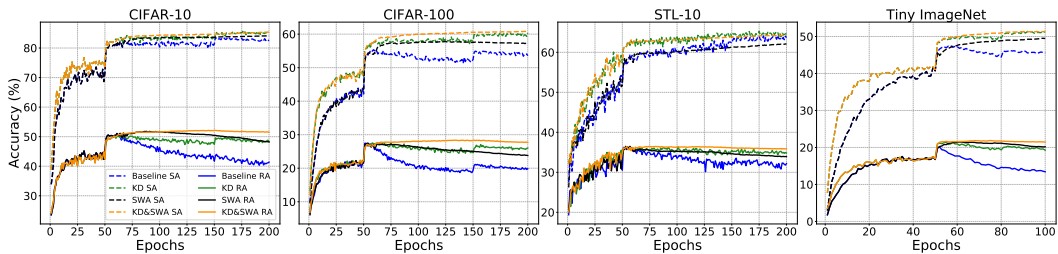

Figure 3: Results of testing accuracy over epochs for ResNet-18 trained on CIFAR-10, CIFAR-100, STL-10, and Tiny ImageNet. Dash / Solid lines show the standard accuracy (SA) / robust accuracy (RA). Blue, Green, **Black** and Orange curves represent the performance of Baseline, KD, SWA and KD&SWA respectively.

where T indexes the training epoch, $n$ the number of past checkpoints to be averaged, $\mathcal{W}_{\mathrm{SWA}}$ the averaged network weight, $\mathcal{W}$ the current network weight, and $\Delta\mathcal{W}$ the SGD update.

## 3  EXPERIMENT AND ANALYSIS

**Datasets**   We consider five datasets in our experiments: CIFAR-10, CIFAR-100 (Krizhevsky & Hinton, 2009), SVHN (Netzer et al., 2011), STL-10 (Coates et al., 2011) and Tiny-ImageNet (Deng et al., 2009). In all experiments, we randomly split the original training set into one training and one validation sets with a 9:1 ratio. Due to the limited space, we place the SVHN results in Appendix A1.3. The ablation studies and the visualizations are mainly on CIFAR-10 and CIFAR-100.

**Attack Methods**   We consider three representative attacks: FGSM (Goodfellow et al., 2015), PGD (Madry et al., 2018), and TRADES (Zhang et al., 2019b). All of them are applied with ($\ell_2,\epsilon = 128/255$) or ($\ell_\infty,\epsilon = 8/255$) setting as in (Madry et al., 2018), to generate adversarial samples. We use FSGM-1/PGD-10/TRADES-10 for training and PGD-20 for testing as the default setting, following Madry et al. (2018); Chen et al. (2020a). In addition, we use Auto-Attack (Croce & Hein, 2020) and CW Attack (Carlini & Wagner, 2017) for a more rigorous evaluation. More details are provided in the Appendix A2.2.

**Training and Evaluation Details**   For all experiments, we by default use ResNet-18 (He et al., 2016), with the exception of VGG-16 (Simonyan & Zisserman, 2014) and Wide-ResNet (Zagoruyko & Komodakis, 2016) adopted in Table 3. For training, we adopt an SGD optimizer with a momentum of 0.9 and weight decay of $5 \times 10^{-4}$, for a total of 200 epochs, with a batch size of 128. The learning rate starts from 0.1 (0.01 for SVHN (Rice et al., 2020)), decay to one-tenth at epochs 50 and 150 respectively. For Tiny-ImageNet, we train for 100 epochs, and the learning rate decay at epochs 50 and 80 with other settings unchanged. The self-training KD regularization is applied throughout the entire training, and SWA is employed after the first learning rate decay (when the robust overfitting usually starts to occur). We evaluate two common metrics that are widely adopted (Zhang et al., 2019b; Chen et al., 2020a): Standard Testing Accuracy (**SA**), and Robust Testing Accuracy (**RA**), which are the classification accuracies on the original and the attacked test sets, respectively.

### 3.1  TACKLING ROBUST OVERFITTING

**Superior Performance Across Datasets**   Table 1 demonstrates our proposal on STL-10, CIFAR-10, CIFAR-100, and Tiny-ImageNet. We consider PGD-AT (Madry et al., 2018) as *Baseline*; and denote our two training techniques as $+\mathcal{KD}_{\mathrm{std\&adv}}$ (KD with standard and robust self-teachers), and *+SWA*, respectively. To numerically show the gap of robust overfitting, we also report the *best* RA values when early stopping during training, the *final* RA in the last epoch, and as the *difference* between *final* minus *best*. For reference, we also report the corresponding SA for the same best-RA checkpoint (not the best SA value throughout training), the final epoch SA, and their difference.

We first observe that the robust overfitting prevails in all *Baseline* cases, with RA differences between final and best early-stopping values as large as 9.34% (CIFAR-10). In comparison, SA stays stable (with negative gaps on STL-10 and CIFAR-10) or continues to improve along with more training epochs (with small positive gaps on CIFAR-100 and Tiny-ImageNet). Fortunately, the gaps were significantly reduced by $+\mathcal{KD}_{\mathrm{std\&adv}}$; and further diminished to only 0.4% to 0.6% when SWA is also applied.

Table 1: Performance showing the occurrence of robust overfitting across datasets and the effectiveness of our proposed remedies with ResNet-18. The difference between best and final robust accuracy indicates degradation in performance during training. We pick the checkpoint which has the **best robust accuracy** on the validation set. The best results and the smallest performance differences are marked in bold.

| Dataset | Settings | Robust Accuracy (RA) | | | Standard Accuracy (SA) | | |
| | | Best | Final | Diff. | Best | Final | Diff. |
| --- | --- | --- | --- | --- | --- | --- | --- |
| STL-10 | Baseline | 36.24 | 32.20 | 4.04 | 56.50 | 63.69 | -7.19 |
| | Baseline + $\mathcal{KD}_{\mathrm{std\&adv}}$ | 36.39 | 34.83 | 1.56 | 62.44 | **64.81** | -2.37 |
| | Baseline + $\mathcal{KD}_{\mathrm{std\&adv}}$ + SWA | **36.46** | **35.90** | **0.56** | **62.76** | 64.24 | **-1.48** |
| CIFAR-10 | Baseline | 50.72 | 41.38 | 9.34 | 80.78 | 82.44 | -1.66 |
| | Baseline + $\mathcal{KD}_{\mathrm{std\&adv}}$ | 50.89 | 48.26 | 2.63 | 83.67 | 85.25 | -1.58 |
| | Baseline + $\mathcal{KD}_{\mathrm{std\&adv}}$ + SWA | **52.14** | **51.53** | **0.61** | **84.65** | **85.40** | **-0.75** |
| CIFAR-100 | Baseline | 27.32 | 19.84 | 7.48 | 53.90 | 53.56 | 0.34 |
| | Baseline + $\mathcal{KD}_{\mathrm{std\&adv}}$ | 27.56 | 26.02 | 1.54 | 57.42 | 60.34 | -2.92 |
| | Baseline + $\mathcal{KD}_{\mathrm{std\&adv}}$ + SWA | **28.28** | **27.69** | **0.59** | **60.58** | **60.85** | **-0.27** |
| Tiny-ImageNet | Baseline | 19.81 | 13.43 | 6.38 | 45.85 | 45.58 | **0.27** |
| | Baseline + $\mathcal{KD}_{\mathrm{std\&adv}}$ | 21.45 | 19.25 | 2.20 | 48.98 | 51.15 | -2.17 |
| | Baseline + $\mathcal{KD}_{\mathrm{std\&adv}}$ + SWA | **21.84** | **21.45** | **0.39** | **50.57** | **51.38** | -0.81 |

Further, we observe our methods to push the best RA higher by $0.22\% \sim 2.03\%$. For example, the best RA on Tiny-ImageNet rises from $19.81\%$ to $21.84\%$. Meanwhile, since there is no longer robust overfitting early in training, the best RA checkpoints become to select late epochs (often close to the end). Consequently, the SA values of the selected best RA models are all substantially improved. For example on CIFAR-100, the standard accuracy of our methods (best RA checkpoint) surpasses the baseline's best RA checkpoint by $6.68\%$, and by $7.29\%$ for the final checkpoint.

Figure 3 further plots the RA and SA curves during training, from which we can clearly observe the diminishing of robust overfitting, after applying $\mathcal{KD}_{\mathrm{std\&adv}}$, *SWA* and a combination of two methods. The training curves robustly improve until the end, without compromising the best achievable RA results, and further leads to a much-improved trade-off between RA and SA by avoiding early stopping (e.g., selecting an early checkpoint for RA, when SA might still be half-baked).

**Across Perturbations and Robustified Methods** Our success can extend beyond PGD-AT. Table 2 presents more results in different perturbations (i.e. $\ell_2$, $\ell_\infty$) and diverse robustified methods (i.e. FSGM in (Wong et al., 2020), TRADES in (Zhang et al., 2019b)). Consistent observations can be made: almost eliminated robust overfitting gaps, and significant gains on RA (by $0.61\% \sim 3.11\%$) and SA (by $1.80\% \sim 4.22\%$).

We also compare with previous state-of-the-art results in (Rice et al., 2020) under the same setting. As shown in Table A7 (Appendix), our methods shrink the gap between the RA best checkpoint and the final epoch RA from $5.70\%$ to $0.17\%$ and simultaneously improve $4.50\%$ by RA and $3.04\%$ by SA. More results can be found in Appendix A1.

**Across Architectures and Improved Attacks** Table 3 demonstrates the effectiveness of our methods across different architectures, including VGG-16, Wide-ResNet-34-4, and Wide-ResNet-34-10. Specifically, our methods reduce the drop of robust accuracy from $5.83\%$ to $0.06\%$ with VGG-16 on CIFAR-10 while achieving an extra robust accuracy improvement of $2.57\%$, $1.69\%$ and $1.23\%$ with VGG-16, Wide-ResNet-34-4 and Wide-ResNet-34-10 on CIFAR10, respectively. To further verify the improvements achieved by our methods, we conduct extra evaluations under improved attacks. As shown in Table 4, after applying the combination of KD and SWA, the overfitting problem is largely mitigated under both Auto-Attack (Croce & Hein, 2020) and CW attack (Carlini & Wagner, 2017). Take CIFAR-10 $\ell_\infty$ adversary as an example, our approaches shrink the drop of robust accuracy from $7.04\%$ to $-0.09\%$ under Auto-Attack, and $14.96\%$ to $0.79\%$ under CW attack, when comparing the best model to the eventually converged model. These results indicate that our methods can generalize to different architectures and improved attacks.

**Excluding Obfuscated Gradients** An often argued "counterfeit" of improved robustness is caused by less effectiveness of generated adversarial examples due to obfuscated gradients (Athalye et al., 2018). To exclude this possibility, we show that our methods maintain improved robustness under unseen transfer attacks. To start with, the left figure in Figure 4 shows the transfer testing performance on an unseen robust model (here we use a separately robustified ResNet-50 with PGD-10

Table 2: Controlled experiments on CIFAR-10. The difference between best and final robust accuracy indicates degradation in performance during training. We pick the checkpoint which has the **best robust accuracy** on the validation set. The best results and the smallest performance difference are marked in bold.

| Adversary | Norm | Radius | Settings | Robust Accuracy (RA) | | | Standard Accuracy (SA) | | |
|---|---|---|---|---|---|---|---|---|---|
| | | | | Best | Final | Diff. | Best | Final | Diff. |
| PGD | $\ell_2$ | $\epsilon = \frac{128}{255}$ | Baseline | 67.26 | 65.93 | 1.33 | 88.78 | 88.86 | **-0.08** |
| | | | Our Methods | **70.37** | **70.23** | **0.14** | **90.58** | **90.48** | 0.10 |
| FGSM | $\ell_2$ | $\epsilon = \frac{128}{255}$ | Baseline | 66.48 | 64.12 | 2.36 | 88.77 | 89.40 | -0.63 |
| | | | Our Methods | **69.53** | **69.49** | **0.04** | **91.11** | **91.15** | **-0.04** |
| FGSM | $\ell_\infty$ | $\epsilon = \frac{8}{255}$ | Baseline | 43.14 | 37.09 | 6.05 | 85.77 | 86.51 | -0.74 |
| | | | Our Methods | **43.75** | **43.74** | **0.01** | **88.53** | **88.72** | **-0.19** |
| TRADES | $\ell_2$ | $\epsilon = \frac{128}{255}$ | Baseline | 69.07 | 64.81 | 4.26 | 85.43 | 85.61 | -0.18 |
| | | | Our Methods | **70.30** | **69.60** | **0.70** | **89.65** | **89.55** | 0.10 |
| TRADES | $\ell_\infty$ | $\epsilon = \frac{8}{255}$ | Baseline | 51.07 | 47.32 | 3.75 | 79.67 | 81.81 | -2.14 |
| | | | Our Methods | **52.92** | **52.28** | **0.64** | **82.95** | **83.31** | **-0.36** |

Table 3: Controlled experiments across different architecture on CIFAR-10/100 under $\ell_\infty$ adversary. The difference between best and final robust accuracy indicates degradation in performance during training. We pick the checkpoint which has the **best robust accuracy** on the validation set. The best results and the smallest performance difference are marked in bold.

| Architecture | Dataset | Settings | Robust Accuracy (RA) | | | Standard Accuracy (SA) | | |
|---|---|---|---|---|---|---|---|---|
| | | | Best | Final | Diff. | Best | Final | Diff. |
| VGG-16 | CIFAR-10 | Baseline | 46.42 | 40.59 | 5.83 | 75.29 | 79.54 | -4.25 |
| | | Our Methods | **48.99** | **48.93** | **0.06** | **79.00** | **79.69** | **-0.69** |
| VGG-16 | CIFAR-100 | Baseline | 21.64 | 17.43 | 4.21 | 39.26 | 45.84 | -6.58 |
| | | Our Methods | **24.79** | **24.73** | **0.06** | **48.20** | **49.00** | **-0.80** |
| WRN-34-4 | CIFAR-10 | Baseline | 52.59 | 43.06 | 9.53 | 81.53 | 83.28 | -1.75 |
| | | Our Methods | **54.28** | **53.90** | **0.38** | **85.17** | **85.50** | **-0.33** |
| WRN-34-4 | CIFAR-100 | Baseline | 28.02 | 20.61 | 7.41 | 53.19 | 53.63 | **-0.44** |
| | | Our Methods | **30.10** | **29.80** | **0.30** | **57.23** | **58.05** | -0.82 |
| WRN-34-10 | CIFAR-10 | Baseline | 54.27 | 47.12 | 7.15 | 84.16 | 85.72 | -1.56 |
| | | Our Methods | **55.50** | **55.34** | **0.16** | **86.81** | **87.06** | **-0.25** |
| WRN-34-10 | CIFAR-100 | Baseline | 29.95 | 24.02 | 5.93 | 56.56 | 56.42 | **0.14** |
| | | Our Methods | **31.93** | **31.51** | **0.42** | **60.86** | **61.78** | -0.92 |

Table 4: Evaluation under improved attacks on CIFAR-10/100 with ResNet-18. The difference between best and final robust accuracy indicates degradation in performance during training. We pick the checkpoint which has the **best robust accuracy** under PGD-20 attack on the validation set. The best results and the smallest performance difference are marked in bold.

| Dataset | Norm | Radius | Settings | Auto-Attack | | | CW Attack | | |
|---|---|---|---|---|---|---|---|---|---|
| | | | | Best | Final | Diff. | Best | Final | Diff. |
| CIFAR-10 | $\ell_2$ | $\epsilon = \frac{128}{255}$ | Baseline | 67.18 | 64.29 | 2.89 | 73.80 | 53.77 | 20.03 |
| | | | Our Methods | **68.87** | **68.90** | **-0.03** | **73.89** | **73.79** | **0.10** |
| CIFAR-10 | $\ell_\infty$ | $\epsilon = \frac{8}{255}$ | Baseline | 47.00 | 39.96 | 7.04 | 75.48 | 60.52 | 14.96 |
| | | | Our Methods | **49.35** | **49.44** | **-0.09** | **77.83** | **77.04** | **0.79** |
| CIFAR-100 | $\ell_2$ | $\epsilon = \frac{128}{255}$ | Baseline | 37.16 | 33.43 | 3.73 | 48.43 | 37.73 | 10.70 |
| | | | Our Methods | **40.56** | **40.61** | **-0.05** | **51.02** | **50.90** | **0.12** |
| CIFAR-100 | $\ell_\infty$ | $\epsilon = \frac{8}{255}$ | Baseline | 22.73 | 18.11 | 4.62 | 45.89 | 37.76 | 8.13 |
| | | | Our Methods | **25.42** | **25.35** | **0.07** | **49.46** | **49.07** | **0.39** |

on CIFAR-100), using attacks generated by the different epoch checkpoints of PGD-AT Baseline, Baseline + $\mathcal{KD}_{\text{std\&adv}}$, and Baseline + $\mathcal{KD}_{\text{std\&adv}}$ + SWA. A higher robust accuracy on the unseen robust model corresponds to a weaker attack. Apparently, our methods consistently yield stronger and more transferable attacks, while the attacking quality generated by the baseline quickly drops with deteriorated transferability. Similarly, the right figure of Figure 4 transfers the attack from

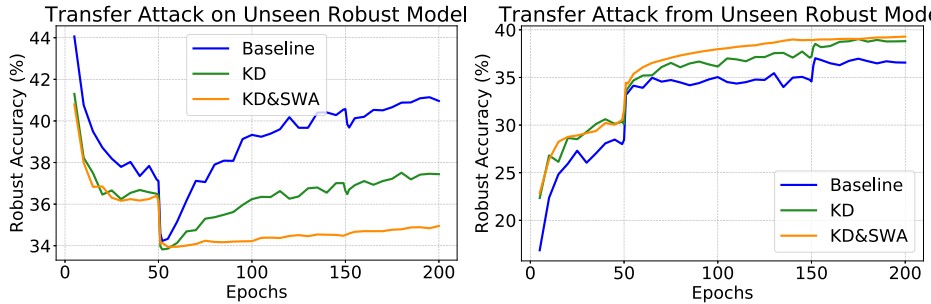

Figure 4: (Left) Transfer attack performance on an unseen robust model, where attacks are generated by Baseline, KD, and KD&SWA's different epoch checkpoints. (Right) Transfer attack performance on models from Baseline, KD, and KD&SWA, where attacks are generated by an unseen robust model. The unseen robust model is a ResNet-50 trained by PGD-10. All experiments are conducted on the CIFAR-100 dataset.

Table 5: Ablation studies on CIFAR-10 with ResNet-18. Compared with Baseline (PGD-AT) methods, the performance improvements and degradations by adding each component are reported in red and blue numbers.

| Settings | Robust Accuracy (RA) | | | Standard Accuracy (SA) | | |
|---|---|---|---|---|---|---|
| | Best | Final | Diff. | Best | Final | Diff. |
| Baseline | 50.72 | 41.38 | 9.34 | 80.78 | 82.44 | -1.66 |
| $\mathcal{KD}_{std}$ | 48.72(↓ 2.00) | 43.08(↑ 1.70) | 5.64 | 83.40(↑ 2.62) | 84.80(↑ 2.36) | -1.40 |
| $\mathcal{KD}_{adv}$ | 51.01 (↑ 0.29) | 47.72(↑ 6.34) | 3.29 | 82.18(↑ 1.40) | 83.51(↑ 1.07) | -1.33 |
| SWA | 51.65(↑ 0.93) | 48.21(↑ 6.83) | 3.44 | 83.42(↑ 2.64) | 84.12(↑ 1.68) | **-0.70** |
| $\mathcal{KD}_{std\&adv}$ | 50.89(↑ 0.17) | 48.26(↑ 6.88) | 2.63 | 83.67(↑ 2.89) | 85.25(↑ 2.81) | -1.58 |
| $\mathcal{KD}_{std\&adv}$ + SWA | **52.14**(↑ 1.42) | **51.53**(↑ 10.15) | **0.61** | **84.65**(↑ 3.87) | **85.40**(↑ 2.96) | -0.75 |

an unseen robust model to the above three methods, while our methods consistently defend better. Those empirical pieces of evidence suggest that our RA gains are not a result of gradient masking.

## 3.2 ABLATION STUDY AND VISUALIZATION

$\mathcal{KD}_{adv}$, $\mathcal{KD}_{std}$ **and SWA** We study the effectiveness of each component in logit and weight smoothening. We also specifically decompose $\mathcal{KD}_{std\&adv}$ into two ablation methods: $\mathcal{KD}_{std}$ (by setting $\lambda_2 = 0$ in Eqn. (1)), and $\mathcal{KD}_{adv}$ (by setting $\lambda_1 = 0$), respectively. Table 5 shows that $\mathcal{KD}_{std}$, $\mathcal{KD}_{adv}$ and SWA all substantially contribute to suppressing the robust overfitting and enhancing the SA-RA trade-off. We notice that while $\mathcal{KD}_{std}$ seems to (understandably) sacrifice the best RA a bit for improving TA, combining it with $\mathcal{KD}_{adv}$ brings the RA compromise back and boosts them both.

**Naive LS versus learned logit smoothening** As $\mathcal{KD}$ could be viewed as a learned version of LS (Yuan et al., 2020), we next quantify the benefit of using $\mathcal{KD}_{std\&adv}$, compared to naive LS (Szegedy et al., 2016), and the teacher-free knowledge distillation regularization (Tf-KD$_{reg}$) in (Yuan et al., 2020), all incorporated with PGD-AT on CIFAR-10. Table 6 show that

Table 6: Ablation of label smoothing versus $\mathcal{KD}$ on CIFAR-10.

| Settings | Robust Accuracy (RA) | | | Standard Accuracy (SA) | | |
|---|---|---|---|---|---|---|
| | Best | Final | Diff. | Best | Final | Diff. |
| Label Smoothing | 50.75 | 44.73 | 6.02 | 81.28 | 83.54 | -2.26 |
| Tf-KD$_{reg}$ | 50.79 | 45.65 | 5.14 | 79.63 | 83.56 | -3.93 |
| $\mathcal{KD}_{std\&adv-FGSM}$ | 50.81 | 48.32 | 2.49 | 84.79 | 85.59 | -0.80 |
| $\mathcal{KD}_{std\&adv-PGD10}$ | 50.89 | 48.26 | 2.63 | 83.67 | 85.25 | -1.58 |
| $\mathcal{KD}_{std\&adv-PGD100}$ | 50.92 | 48.47 | 2.45 | 83.31 | 84.64 | -1.33 |

both naive LS and Tf-KD$_{reg}$ also reduce robust overfitting to some extent, but far less competitive than $\mathcal{KD}_{std\&adv}$. Moreover, the robustness gains of naive LS and Tf-KD$_{reg}$ no longer hold under transfer attacks, implying that they are susceptible to obfuscated gradients. Further visualization in Figure 5 demonstrates that our methods smooth the logits without compromising the class-wise discriminative information, while naive LS and Tf-KD might suffer from weaker gradients here.

**Quality of Self-Teachers** An extra price for our learned logit smoothening is the pre-training of self-teachers, although this is already quite common in similar literature (Chen et al., 2020a;b). To further reduce this burden, we explore whether high-quality and more expensive pre-training is necessary for us, and fortunately find that is not the case. For example, Table 6 shows only marginal performance difference when the robust self-teacher is pre-trained using FGSM or PGD-10/100.

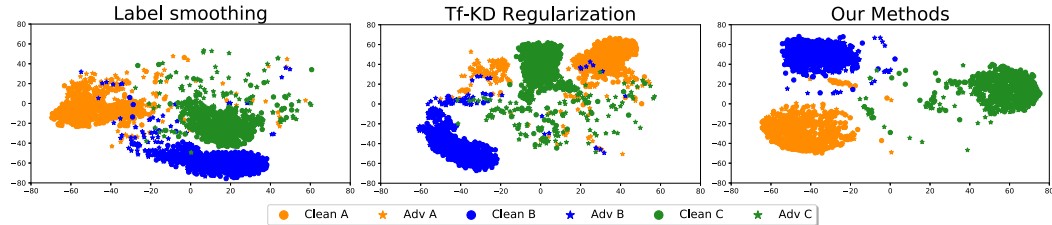

Figure 5: t-SNE results of different logits smoothing approaches on CIFAR-10. Dots and stars represent for clean and adversarial images respectively. Orange, Blue and Green represent classes A, B and C respectively. For each class, we visualize all **testing images** and their corresponding adversarial images from PGD-20.

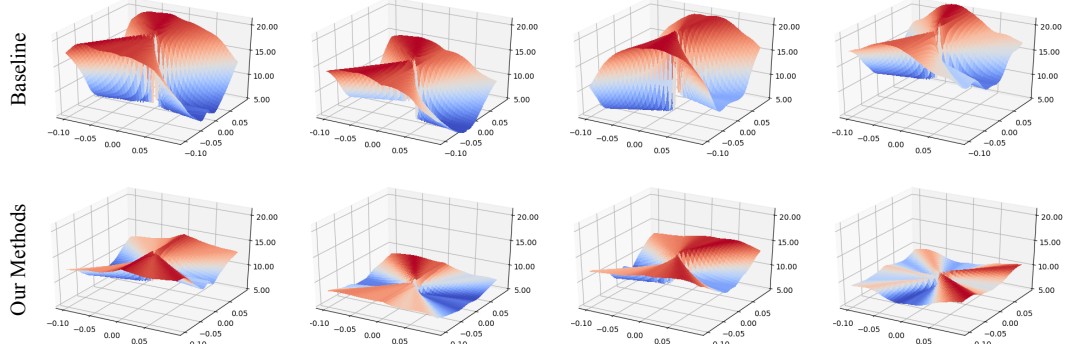

Figure 6: Comparison of loss landscapes of models trained by baseline (the first row) and our methods (the second row). Loss plots in each column are generated from the same original image randomly chosen from the CIFAR-100 test dataset. $z$ axis denotes the loss value. Following the settings in (Engstrom et al., 2018), we plot the loss landscape function: $z = \text{loss}(x \cdot r_1 + y \cdot r_2)$, where $r_1 = \text{sign}(\nabla_i f(i))$ and $r_2 \sim \text{Rademacher}(0.5)$. Here $i$ denostes the original image, and $f(\cdot)$ a trained model whose inputs are scaled to $[0, 1]$.

**Visualizing Flatness and Local Linearity** We expect SWA to find flatter minima for AT to improve its generalization, and we show it to indeed happen by visualizing the loss landscape w.r.t. both input and weight spaces. Figure 6 shows that our methods notably flatten the rugged landscape w.r.t. the input space, compared to the PGD-AT baseline, which aligns with the robust generalization claims in (Moosavi-Dezfooli et al., 2019; Wu et al., 2020a). Figure 7 follows (Izmailov et al., 2018) to perturb the trained model in the weight space and show how the robust testing loss changes over the perturbation radius. We perturb 10 different random directions at each different $\ell_2$ distance. Our methods present better weight smoothness around the achieved local minima too, which suggests improved generalization (Dinh et al., 2017; Petzka et al., 2019).

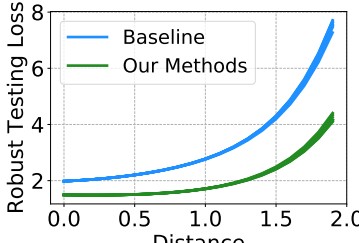

Figure 7: Robust testing loss as a function of perturbed weight distance, starting from models trained by PGD-AT baseline and our methods on CIFAR-100.

We additionally look at the *local linearity* measurement proposed in (Andriushchenko & Flammarion, 2020), which originally addresses catastrophic overfitting in fast AT. As shown in Figure A11, our methods also achieve consistently better local linearity.

## 4 CONCLUSION

This paper takes one more step towards addressing the recently discovered robust overfitting issue in AT. We present two empirical solutions to smooth the logits and weights respectively; both are motivated by successful practice in improving standard generalization, and we adapt them for AT. While Rice et al. (2020) found simpler regularizations unable to fix robustness overfitting, our learned smoothening regularization seems to largely mitigate that. Extensive experiments show our proposal to establish new state-of-the-art performance on AT. While promising progress has been made, the underlying cause of robust overfitting is not yet fully explained. Our future work will connect to more theoretical understandings of this issue (Wang et al., 2019; 2020b).

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

## A1    MORE EXPERIMENT RESULTS

### A1.1    STATE-OF-THE-ART BENCHMARK ON CIFAR-100

We implement our methods with exactly the same setting as (Rice et al., 2020) and compare it with the baseline result reported from the original paper. As shown in Table A7 and Figure A8, our methods achieve great improvements both on robust accuracy and standard accuracy (1.64% in RA and 3.78% in SA for $\ell_\infty$, 4.50% in RA and 3.04% in SA for $\ell_2$), which establish a new state-of-the-art bar.

Table A7: Comparative Experiment on CIFAR-100, we follow the same setting and compare with the baseline result from (Rice et al., 2020). **Best** refers to the model with best robust accuracy during training and **Final** is an average of accuracy over last 5 epochs.

| Adversary | Norm | Radius | Settings | Robust Accuracy (RA) | | | Standard Accuracy (SA) | | |
|---|---|---|---|---|---|---|---|---|---|
| | | | | Best | Final | Diff. | Best | Final | Diff. |
| PGD | $\ell_2$ | $\epsilon = \frac{128}{255}$ | Baseline | 43.20 | 37.50±0.09 | 5.70 | 62.50 | 60.10±0.22 | 2.40 |
| | | | Our Methods | **47.70** | **47.53±0.03** | **0.17** | **65.54** | **65.56±0.01** | **-0.02** |
| PGD | $\ell_\infty$ | $\epsilon = \frac{8}{255}$ | Baseline | 28.10 | 21.40±0.39 | 6.70 | 52.70 | 54.10±0.23 | -1.40 |
| | | | Our Methods | **29.74** | **29.40±0.02** | **0.34** | **56.48** | **57.69±0.03** | **-1.21** |

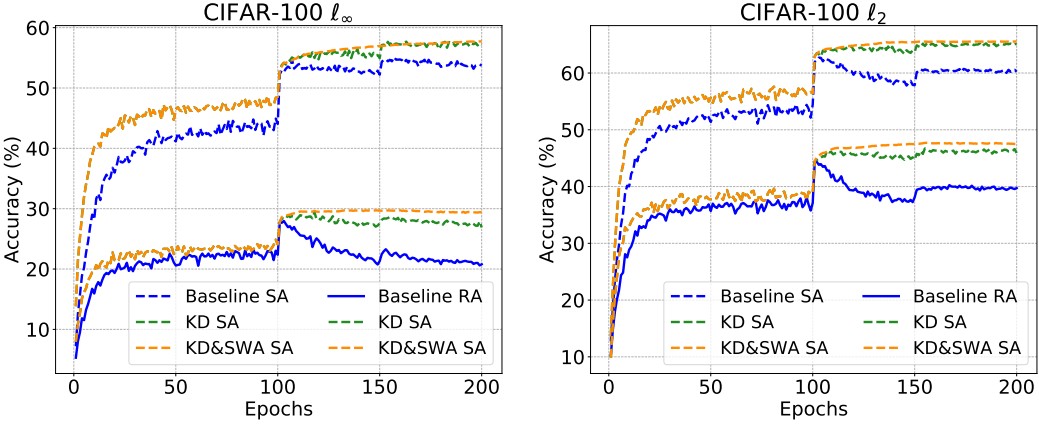

Figure A8: Results of testing accuracy over epochs for ResNet-18 trained on CIFAR-100 with the same setting as Rice et al. (2020). Dash lines show the standard accuracy (SA); solid lines represent the robust accuracy (RA). Blue, Green and Orange curves represent the performance of Baseline, KD and KD&SWA respectively.

### A1.2    T-SNE RESULT ON CIFAR-100

We visualize the learned feature space with all training images and their corresponding adversarial images from PGD-10 on CIFAR-100. As shown in Figure A9, our learned features have a larger distance between classes while being more clustered within the same class. The more distinguishable feature embedding justifies the improvement of both robust and standard accuracy.

### A1.3    SUPERIOR PERFORMANCE ON SVHN

We conduct our experiments on SVHN with ResNet-18 (He et al., 2016) architecture and adopt an SGD optimizer with a momentum of 0.9 and a weight decay of $5 \times 10^{-4}$ for 80 epochs in total with a batch size of 128. The learning rate starts from 0.01 and follows a cosine annealing schedule. The result can be found on Table A8 and Figure A10. As we can see, the robust accuracy of the best checkpoint for $\ell_\infty$ is improved from 52.60% to 53.65%, and robust overfitting is alleviated by 6.30%. In the meantime, standard accuracy has also been improved by 2.47%. The superior performance on SVHN aligns with results on other datasets, which shows the effectiveness of our methods.

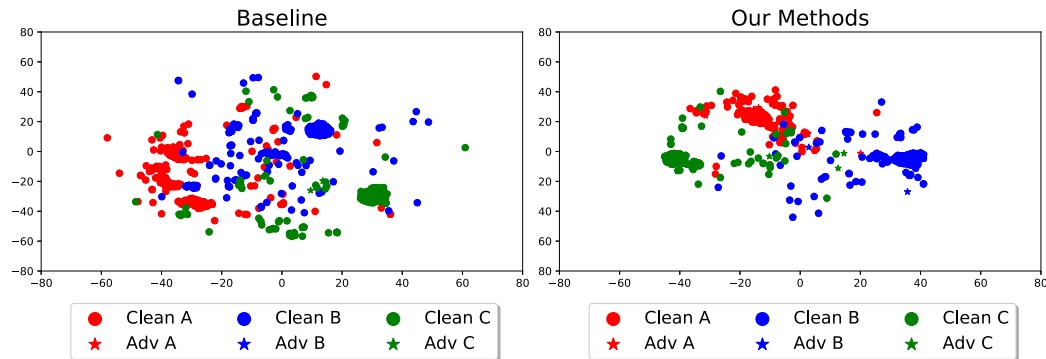

Figure A9: t-SNE results of different models trained on CIFAR-100. Dots and stars represent for clean and adversarial images respectively. Red, Blue and Green represent classes A, B and C respectively. For each class, we visualize all **training images** and their corresponding adversarial images from PGD-10. The left figure is Baseline; the right figure is Our Methods.

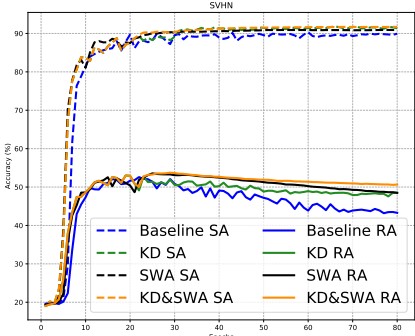

Figure A10: Results of testing accuracy over epochs for ResNet-18 trained on SVHN. Dash lines show the standard accuracy (SA); solid lines represent the robust accuracy (RA). Blue, Green, Black and Orange curves represent the performance of Baseline, KD, SWA and KD&SWA respectively.

Table A8: Performance showing the occurrence of robust overfitting and effectiveness of our proposed remedies with ResNet-18 on SVHN. The difference between best and final robust accuracy indicates degradation in performance during training. We pick the checkpoint which has the best robust accuracy on validation dataset. The best results and the minimum performance difference are marked in bold.

| Dataset | Settings | Robust Accuracy (RA) | | | Standard Accuracy (SA) | | |
|---|---|---|---|---|---|---|---|
| | | Best | Final | Diff. | Best | Final | Diff. |
| | Baseline | 52.60 | 43.30 | 9.30 | 87.93 | 89.94 | -2.01 |
| SVHN | Baseline + $\mathcal{KD}_{\mathrm{std\&adv}}$ | 52.93 | 48.46 | 4.47 | 87.62 | 91.36 | -3.74 |
| | Baseline + $\mathcal{KD}_{\mathrm{std\&adv}}$ + SWA | **53.65** | **50.65** | **3.00** | **90.40** | **91.70** | **-1.30** |

### A1.4 LOCAL LINEARITY

As proposed by (Andriushchenko & Flammarion, 2020), the catastrophic overfitting problem is mainly due to the local linearity reduction when adversarial training with FGSM(Rice et al., 2020). So we borrow this measurement in the robust overfitting scenario, which calculates the expectation of the cosine similarity of the gradient between the original input and randomly perturbed one with a uniform distribution, as shown in Eqn. 3. The result shown in figure A11 indicates that our methods help to slow the decline of local linearity and the maintenance of local linearity is also helpful for preventing robust overfitting.

$$\mathbb{E}_{(\mathbf{x},y)\in\mathcal{D},\boldsymbol{\eta}\in\mathcal{U}([-\epsilon,\epsilon]^d)}\left[\cos\left(\nabla_{\mathbf{x}}\mathcal{L}(f(\boldsymbol{\theta},\mathbf{x}),y),\nabla_{\mathbf{x}}\mathcal{L}(f(\boldsymbol{\theta},\mathbf{x}+\boldsymbol{\eta}),y)\right)\right] \tag{3}$$

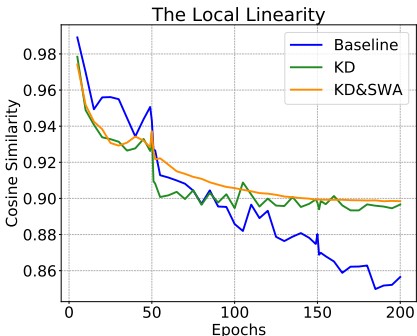

Figure A11: The local linearity is calculated from all test images on CIFAR-100 with models from each period of the training process. Blue, Green and Orange curves represent the local linearity of Baseline, KD and KD&SWA respectively.

## A1.5 Ablation of Transfer attack

With the purpose of fully comparing the effects of label smoothing and knowledge distillation, we introduce a transfer attack with an unseen non-robust model with the same architecture, follow the same setting as (Fu et al., 2020). A higher accuracy on the unseen model indicates a weaker attack generated by the corresponding setting while a higher accuracy from the unseen model means better robustness. As shown in Table A9, only knowledge distillation shows significant improvement with both accuracies, compared with baseline(PGD-AT) methods. The strength of generated adversarial images is improved by 4.32% and the robustness is improved by 2.91% for the best model. We also experiment with an unseen robust model and get consistent improvement. This improvement indicates that knowledge distillation introduces more discriminating information from teacher models, which is better than manually designed label smoothing methods.

Table A9: Ablation of Transfer attack. The accuracy on unseen model is the accuracy of unseen model with adversarial images generated by source models from different settings and the accuracy from unseen model means the opposite. We generated adversarial images for all test images on CIFAR-10 with $\ell_\infty$ PGD-20. Baseline represents the PGD-AT methods.

| Settings | Accuracy on unseen model | | Accuracy from unseen model | |
|---|---|---|---|---|
| | Best | Final | Best | Final |
| Baseline | 69.87 | 80.43 | 79.72 | 81.77 |
| Label smoothing | 70.94 | 81.24 | 78.44 | 82.82 |
| Tf-$\mathcal{KD}_{\mathrm{reg}}$ | 73.29 | 82.21 | 79.47 | 82.71 |
| $\mathcal{KD}_{\mathrm{std\&adv-PGD10}}$ | **65.55** | **72.63** | **82.63** | **84.01** |

## A1.6 SWA versus iSWA

One possible extension of SWA is to replace $\mathcal{W}^{\mathrm{T}-1}$ with $\mathcal{W}_{\mathrm{SWA}}^{\mathrm{T}-1}$ in Eqn. 2. We name this variant as iSWA and compare it with the original SWA in Table A10. Both weight smoothing techniques can mitigate robust overfitting, and iSWA performs slightly better on RA while sacrificing some SA.

Table A10: Ablation of SWA on CIFAR-10.**Best** refers to the model selected with best robust accuracy on validation dataset and **Final** is the model at the end of training process.

| Settings | Robust Accuracy (RA) | | | Standard Accuracy (SA) | | |
|---|---|---|---|---|---|---|
| | Best | Final | Diff. | Best | Final | Diff. |
| $\mathcal{KD}_{\text{std\&adv}-\text{PGD10}}$ + SWA | 52.14 | 51.53 | 0.61 | 84.65 | 85.40 | -0.75 |
| $\mathcal{KD}_{\text{std\&adv}-\text{PGD10}}$ + iSWA | 52.36 | 52.33 | 0.03 | 83.17 | 83.54 | -0.37 |

## A2 MORE METHODOLOGY AND IMPLEMENTATION DETAILS

### A2.1 KNOWLEDGE DISTILLATION

We state $\mathcal{KD}$ as follows:

$$\mathcal{KD}(\boldsymbol{y}, \hat{\boldsymbol{y}}) = -\mathcal{H}(t(\boldsymbol{y}), t(\hat{\boldsymbol{y}}))$$
$$= -\sum_j t(\boldsymbol{y})_j \log t(\hat{\boldsymbol{y}})_j$$

where $t(\boldsymbol{y})_i = \frac{(\boldsymbol{y}_i)^{1/\text{T}}}{\sum_j (\boldsymbol{y}_j)^{1/\text{T}}}$, T $= 2$ in our case, following the standard setting in (Hinton et al., 2015; Li & Hoiem, 2017).

### A2.2 ADVERSARIAL TRAINING

Adversarial training incorporates generated adversarial examples into the training process and significantly improves the robustness of networks. In our paper, we implemented three different adversarial training schemes: FGSM, PGD and TRADES, which can be described as the optimization problem below: Eqn.4 for FGSM and PGD, Eqn.5 for TRADES.

$$\min_{\boldsymbol{\theta}} \mathbb{E}_{(\mathbf{x},y)\in\mathcal{D}} \max_{\boldsymbol{\delta}\in\text{B}_\epsilon(\mathbf{x})} \mathcal{L}(f(\boldsymbol{\theta}, \mathbf{x}+\boldsymbol{\delta}), y) \tag{4}$$

$$\min_{\boldsymbol{\theta}} \mathbb{E}_{(\mathbf{x},y)\in\mathcal{D}} \left[ \mathcal{L}(f(\boldsymbol{\theta}, \mathbf{x}), y) + \beta \cdot \max_{\boldsymbol{\delta}\in\text{B}_\epsilon(\mathbf{x})} \mathcal{KL}(f(\boldsymbol{\theta}, \mathbf{x}+\boldsymbol{\delta}), f(\boldsymbol{\theta}, \mathbf{x})) \right] \tag{5}$$

As for the maximization process, FGSM perturbs the input with a single step in the direction of the sign of the gradient and PGD is the iterative form of FGSM with random restarts, which works as follows.

$$\boldsymbol{\delta}^{t+1} = \text{proj}_{\mathcal{P}}\left(\boldsymbol{\delta}^t + \alpha \cdot \text{sgn}\left(\nabla_{\mathbf{x}} \mathcal{L}(f(\boldsymbol{\theta}, \mathbf{x}+\boldsymbol{\delta}^t), y))\right)\right) \tag{6}$$

$$\boldsymbol{\delta}^{t+1} = \text{proj}_{\mathcal{P}}\left(\boldsymbol{\delta}^t + \alpha \cdot \text{sgn}\left(\nabla_{\mathbf{x}} \mathcal{KL}(f(\boldsymbol{\theta}, \mathbf{x}+\boldsymbol{\delta}), f(\boldsymbol{\theta}, \mathbf{x})))\right)\right) \tag{7}$$

TRADES (Eqn.7) replaces the cross entropy loss in PGD with the Kull-back–Leibler divergence of network output for the clean input and the adversarial input. Where f is the network with parameters $\boldsymbol{\theta}$, $(\mathbf{x}, y)$ is the data. $\alpha$ is the step size and $\boldsymbol{\delta}^t$ is the adversarial perturbation after t times iterations. The perturbation is constrained in an $\ell_p$ norm ball, i.e.$\|\boldsymbol{\delta}\|_p \leq \epsilon$, which is realized by projection. we consider both $\ell_\infty$ and $\ell_2$ in our paper. For $\ell_\infty$ adversary, we use $\epsilon = \frac{8}{255}$ and $\alpha = \frac{2}{255}$ for PGD and TRADES with 10 steps in training and 20 steps in testing, while using $\alpha = \frac{7}{255}$ for FGSM during training. As for $\ell_2$ adversary, we use $\epsilon = \frac{128}{255}$ and $\alpha = \frac{15}{255}$ with the same steps as $\ell_\infty$ adversary in all three attack methods.

For a comprehensive evaluation, we consider two improved attacks, i.e., Auto-Attack (Croce & Hein, 2020) and CW Attack (Carlini & Wagner, 2017). We use the official implementation and default settings for Auto-Attack ($\ell_\infty$ with $\epsilon = \frac{8}{255}$ and $\ell_2$ with $\epsilon = \frac{128}{255}$) and the implementation

from AdverTorch (Ding et al., 2019) for CW attack with the same setting as Rony et al. (2019), specifically, 1 search step on C with an initial constant of 0.1, with 100 iterations for each search step and 0.01 learning rate. Detailed links are provided below:

- The Official Repository: `https://github.com/fra31/auto-attack`
- The Leaderboard: `https://robustbench.github.io/`

