# OpenReview forum: "Robust Overfitting may be mitigated by properly learned smoothening"
_ICLR.cc/2021/Conference — ICLR 2021 Poster_

### Official Review · AnonReviewer2 · 2020-10-27
**An interesting potential approach to prevent robust overfitting**

**Rating:** 6
**Confidence:** 3

**Review:**

The paper studies a method for mitigating robust overfitting. Rice et al., and others have observed that when training a neural network robustly on say CIFAR10, then the robust test error often overfits, i.e., it has a U-shaped curve as a function of training epochs. Rice et al. demonstrated that early stopping the robust training enables state-of-the-art robust performance. However, to realize this performance, it is necessary to find a good early stopping point, which can be difficult (but can be found with testing on a validation set). The paper proposes an alternative to early stopping: smoothing the logits and smoothing the weights, by using  two existing techniques, namely self-training and stochastic weight averaging. The paper finds that smoothing mitigates robust overfitting, and reports even a slight improvement over early stopping at the optimal point.

Strength:
- The paper proposes two existing techniques to prevent robust overfitting:  self-training and stochastic weight averaging, and shows that combining those two approaches is very effective in preventing robust overfitting for ResNet-18.
- The paper is well written and easy to follow.

Weaknesses:
- All experiments are carried out with one network only: ResNet-10. To validate the claim that learned label smoothing can mitigate robust training, is important to test the label smoothing method on a variety of setups and models.
- The improvement are tiny relativ to early stopping, and another submission to this workshop (https://openreview.net/pdf?id=Xb8xvrtB8Ce) has shown that the baseline the paper under review is comparing to (the setting of Rice et al.) is quite brittle relative to choices of hyperparameters such as slight differences in weight decay. Therefore, the gains obtained by the paper under review could very well be subsumed by slightly tuning early stopping.

Summary: It is an important problem to study methods that mitigate robust overfitting, and the paper proposes a combination of two smoothing techniques and demonstrates its effectiveness through extensive experiments. I'm therefore leaning to recommend acceptance of this paper, however, as mentioned, the paper's results might not generalize to other models and the slight gains over early stopping might be void by slightly tuning ES better. Therefore, I would not be upset if this paper were rejected. In any case, the paper's results would be more convincing if it would contain results for different models (e.g., VGG and other deep nets but also simple baseline models such as random feature models), and if it would contain a simple theoretical statement to provide intuition why label smoothing should help.

----
UPDATE: Thanks; I have read the response, kept my score, and responded below.

---

> ### Author Response · Authors · 2020-11-20
> **Response to Reviewer #2 [Cons 1]**
>
> [Cons1: More Setups and Models] Thanks for the helpful suggestions. We have conducted new experiments using VGG16, Wide-ResNet-34-4, and Wide-ResNet-34-10 on the CIFAR-10/100 dataset and report the results in Table S1. And if the reviewer could kindly provide more details or some references about the random feature models, we are very willing to conduct new experiments and report the results. As we can see, our approaches largely mitigate robust overfitting across multiple models. For example, in VGG16, our methods reduce the gap between the RA best checkpoint and the final epochs from 5.83% to 0.06% on CIFAR-10 and 4.21% to 0.06% on CIFAR-100. Meanwhile, our methods gain an extra robustness improvement (2.57% on CIFAR-10 and 3.15% on CIFAR-100) compared with early stopping. Consistent improvements can also be observed with Wide-ResNets. In addition, we conduct more experiments to verify our approaches under Auto-Attack and CW attack, which is shown in Table S2. Evaluated under the more rigorous attack method, i.e. Auto-Attack, our methods can still be effective. Compared with the results reported in the Auto-Attack leaderboard from [2], our approaches reach 1.19% and 6.47% improvement on CIFAR-10 and CIFAR-100 under ResNet18, respectively. Besides, there is no significant drop of robust accuracy (RA) between the RA best checkpoint and the final epoch under both Auto-Attack and CW attack.
>
> [1] Bag of Tricks for adversarial training
>
> [2] Overfitting in adversarially robust deep learning
>
> [3] Decoupling Direction and Norm for Efficient Gradient-Based L2 Adversarial Attacks and Defenses
>
>
> Table S1 Performance showing the occurrence of robust overfitting across different architectures and the effectiveness of our proposed remedies under $\ell_\infty$ PGD-20 adversary. We pick the checkpoint which has the best robust accuracy on the validation set.
>
> |Architecture|Dataset|Settings|Robust Accuray(RA)(Best->Final)| Standard Accuracy(SA)(Best->Final)|
> |:-:|:-:|:-:|:-:|:-:|
> |VGG-16|CIFAR-10|Baseline|(46.42→40.59)(↓5.83)|(75.29→79.54)(↑4.25)|
> |VGG-16|CIFAR-10|Our Methods|(48.99→48.93)(↓0.06)|(79.00→79.69)(↑0.69)|
> |VGG-16|CIFAR-100|Baseline|(21.64→17.43)(↓4.21)|(39.26→45.84)(↑6.58)|
> |VGG-16|CIFAR-100|Our Methods|(24.79→24.73)(↓0.06)|(48.20→49.00)(↑0.80)|
> |WideResNet-34-4|CIFAR-10|Baseline|(52.59→43.06)(↓9.53)|(81.53→83.28)(↑1.75)|
> |WideResNet-34-4|CIFAR-10|Our Methods|(54.28→53.90)(↓0.38)|(85.17→85.50)(↑0.33)|
> |WideResNet-34-4|CIFAR-100|Baseline|(28.02→20.61)(↓7.41)|(53.19→53.63)(↑0.44)|
> |WideResNet-34-4|CIFAR-100|Our Methods|(30.10→29.80)(↓0.30)|(57.23→58.05)(↑0.82)|
> |WideResNet-34-10|CIFAR-10|Baseline|(54.27→47.12)(↓7.15)|(84.16→85.72)(↑1.56)|
> |WideResNet-34-10|CIFAR-10|Our Methods|(55.50→55.34)(↓0.16)|(86.81→87.06)(↑0.25)|
> |WideResNet-34-10|CIFAR-100|Baseline|(29.95→24.02)(↓5.93)|(56.56→56.42)(↓0.14)|
> |WideResNet-34-10|CIFAR-100|Our Methods|(31.93→31.51)(↓0.42)|(60.86→61.78)(↑0.92)|
>
>
> Table S2 Robust accuracy under Auto-Attack and CW Attack on CIFAR-10/100 with ResNet18. The best checkpoint is picked with the best robust accuracy under PGD-20 on the validation set. We follow the same setting as [3] for CW Attack: 1 search step on C with an initial constant of 0.1, with 100 iterations for each search step and learning rate is 0.01.
>
> |Dataset|Norm|Settings|Auto-Attack(Best->Final)|CW Attack(Best->Final)|
> |:-:|:-:|:-:|:-:|:-:|
> |CIFAR-10|$\ell_\infty$|Baseline|(47.00→39.96)(↓7.04)|(75.48→60.52)(↓14.96)|
> |CIFAR-10|$\ell_\infty$|Our Methods|(49.35→49.44)(↑0.09)|(77.83→77.04)(↓0.79)|
> |CIFAR-10|$\ell_2$|Baseline|(67.18→64.29)(↓2.89)|(73.80→53.77)(↓20.03)|
> |CIFAR-10|$\ell_2$|Our Methods|(68.87→68.90)(↑0.03)|(73.89→73.79)(↓0.10)|
> |CIFAR-100|$\ell_\infty$|Baseline|(22.73→18.11)(↓4.62)|(45.89→37.76)(↓8.13)|
> |CIFAR-100|$\ell_\infty$|Our Methods|(25.42→25.35)(↓0.07)|(49.46→49.07)(↓0.39)|
> |CIFAR-100|$\ell_2$|Baseline|(37.16→33.43)(↓3.73)|(48.43→37.73)(↓10.70)|
> |CIFAR-100|$\ell_2$|Our Methods|(40.56→40.61)(↑0.05)|(51.02→50.90)(↓0.12)|

---

> ### Author Response · Authors · 2020-11-20
> **(Continued) Response to Reviewer #2 [Cons 2 & Open Question]**
>
> [Cons2: Hyperparameters Investigations] Thanks for pointing this out. The tricks for adversarial training studied in the under-review paper [1] help a lot. As inspired by this paper, we conduct an ablation study on the choice of weight decay (WD), i.e., the key factor mentioned in [1]. Follow the same settings in [1], we compared the results of baseline PGD-AT and our methods in Table S3. The performance is sensitive to the choice of WD, which is aligned with the conclusion in [1]. With the same WD, our methods can significantly mitigate robust overfitting and meanwhile achieve an extra robust accuracy improvement (0.43%, 0.21%, and 1.42% for WD of 1e-4, 2e-4, 5e-5, respectively). When choosing 5e-4 of WD, both the baseline and our methods reach the best performance.  We do appreciate the work [1] and conducted new experiments to alleviate the reviewer's concerns. However, we count [1] as a concurrent submission and we hope that it should not be used to downgrade our contributions.
>
> [Open Question: Theoretical Intuition of Label Smoothing.] Thanks for this interesting open question. It is fair to note that our main goal is to provide systematic empirical investigations for alleviating the robust overfitting issue [2]. Besides the impressive experiment results, we also show extensive visualizations to justify how and why the robust overfitting happens and be mitigated.
>
> To our best knowledge, most label smoothing works focus on empirical studies [4]. Recently, we noticed that handful of papers [3] start some initial theoretical analyses on minimization problems. [3,4] point out that label smoothing is related to loss-correction techniques [3], and encourages the representations of training examples from the same class to group in tight clusters [4] which is also observed in our paper (Figure 5). However, the theoretical analyses for min-max problems are still missing in the literature. We sincerely appreciate the suggestion and will further explore the theoretical foundations in the future.
>
>
> [1] Bag of Tricks for adversarial training
>
> [2] Overfitting in adversarially robust deep learning
>
> [3] Does Label Smoothing Mitigate Label Noise?
>
> [4] When Does Label Smoothing Help?
>
> Table S3 An ablation study on the choice of weight decay using ResNet-18, the robust accuracy(RA) is evaluated under PGD-20. We pick the checkpoint which has the best robust accuracy on the validation set.
>
> |Weight Decay|Settings|Robust Accuray(RA)(Best->Final)| Standard Accuracy(SA)(Best->Final)|
> |:-:|:-:|:-:|:-:|
> |1e-4|Baseline|(48.77→39.40)(↓9.37)|(81.47→81.39)(↓0.08)|
> |1e-4|Our Methods|(49.20→48.28)(↓0.92)|(82.35→82.96)(↑0.61)|
> |2e-4|Baseline|(50.45→40.22)(↓10.23)|(81.33→81.51)(↑0.18)|
> |2e-4|Our Methods|(50.66→50.18)(↓0.48)|(82.89→83.73)(↑0.84)|
> |5e-4|Baseline|(50.72→41.38)(↓9.34)|(80.78→82.44)(↑1.66)|
> |5e-4|Our Methods|(52.14→51.53)(↓0.61)|(84.65→85.40)(↑0.75)|

---

> > ### Comment · AnonReviewer2 · 2020-11-22
> > **Comment on revision**
> >
> > Thanks for the detailed response! It is encouraging to see that your proposed method performs on par or slightly better than the previous baseline. [1] is counted as a concurrent submission. I also understand that it might be difficult to derive theoretical results, even for a very simple setting. Nevertheless, the critique remains that the paper conducts relatively narrow experiments (only on ResNet-18, and there is little evidence (neither empirical nor theoretical) that the findings generalize beyond the studied ResNet-18 setting to for example VGG, other deep nets, or even simple baseline models such as random feature models.

---

> > > ### Author Response · Authors · 2020-11-22
> > > **Response to Reviewer #2 [More Extra Experiments are in the Other Part of Our Response]**
> > >
> > > Thanks for your prompt reply. In fact, we believe our other part of the response, “Response to Reviewer #2 [Cons 1]” posted on 20 Nov 2020, had already addressed your concerns.
> > >
> > > We have conducted new experiments using VGG16, Wide-ResNet-34-4, and Wide-ResNet-34-10 on the CIFAR-10/100 dataset and report the results in Table S1. For example, in VGG16, our methods reduce the gap between the RA best checkpoint and the final epochs from 5.83% to 0.06% on CIFAR-10 and 4.21% to 0.06% on CIFAR-100. Meanwhile, our methods gain an extra robustness improvement (2.57% on CIFAR-10 and 3.15% on CIFAR-100) compared with early stopping. Consistent improvements can also be observed with Wide-ResNets. In addition, we conduct more experiments to verify our approaches under Auto-Attack and CW attack, which is shown in Table S2. Evaluated under the more rigorous attack method, i.e. Auto-Attack, our methods can still be effective.
> > >
> > > We invite you to take another look at that part and let us know if you have further questions. Thanks!

---

### Official Review · AnonReviewer3 · 2020-10-27
**This paper uses logit smoothing and weight averaging to enhance adversarial training.**

**Rating:** 7
**Confidence:** 5

**Review:**

#########################################################################

Summary:
This paper uses the existing tricks that can enhance the standard training, to show that combining some of those tricks (in this paper, labels/logits smoothing and weight averaging) can improve adversarial training.

#########################################################################

Pros:
1 Compared with existing weight manipulation AT methods, this paper first utilizes stochastic weight averaging (SWA) (averaging multiple checkpoints along the training trajectory) without incurring computational overhead.

2 This paper conducted experiments across four different datasets.

#########################################################################

Cons:
1 The paper’s novelty is marginal. Specifically, first, label/logit smoothing has been demonstrated effective in adversarial training due to the better separation of different classes. For example, to my knowledge, three papers got accepted with the shared philosophy but slightly different techniques/decorations [1, 2, 3]
Second, as the authors mentioned, manipulating model weights is also shown effective [4].
Therefore, this paper's conceptual improvements are marginal.

[1] Metric Learning for Adversarial Robustness, NeurIPS 2019\
[2] Rethinking Softmax Cross-Entropy Loss for Adversarial Robustness, ICLR 2020\
[3] Boosting Adversarial Training with Hypersphere Embedding, NeurIPS 2020\
[4] Revisiting loss landscape for adversarial robustness, NeurIPS 2020

2 This paper hypothesizes “one source of robust overfitting might lie in that the model ‘overfits’ the attacks generated in the early stage of AT and fails to generalize or adapt to the attacks in the late stage.”  It is not clear to me why this hypothesis is valid. Would you explain more justifications?

3. In Figure 3, are there any experimental results for SWA SA and SWA RA?
Besides, more robustness evaluations are needed, e.g., CW attack, AA attack, Guided Adversarial Margin Attack.
More adversarial training on different network structures are needed, e.g., Wide ResNet.

---

> ### Author Response · Authors · 2020-11-18
> **Response to Reviewer #3 [Cons 1-2]**
>
> [Cons 1: Marginal Contribution.] We disagree that the contribution is marginal. As recognized by Reviewer #2, our work for the first time provides principal and effective solutions to mitigate robust overfitting [5] beyond the early stopping. As shown in Table 3, KD or SWA alone helps alleviate robust overfitting, and the combination of KD and SWA  further improves the performance, indicating their supplementary benefits. Technically, we leverage knowledge distillation and self-training to smooth the logits rather than proposing metric learning regularizers [1,2,3]. For the model weights manipulation, we utilize a much simpler yet effective technique, stochastic weight averaging, while [4] resorts to adversarial weight perturbation via a complex min-max optimization. To the best of our knowledge, none of [1-4] was proposed for solving the problem of robust overfitting. Thus, we believe that our differences with [1-4] are significant. However, we also would like to thank the reviewer for pointing out [1-4] and we have included them in the updated paper. Thanks for the suggestions!
>
> [Cons 2: Hypothesize Explanation.] Thanks for the question. Our explanation on “one source of robust overfitting might lie in that the model ‘overfits’ the attacks generated in the early stage of AT and fails to generalize or adapt to the attacks in the late-stage” can be verified by experiment results in Figure 4 (Left). To be more specific, we evaluate the transferability of attacks generated by models at different epoch checkpoints of PGD-AT Baseline, Baseline + KD, and Baseline + KD + SWA against an unseen victim model given by robustified ResNet-50 with PGD-10 on CIFAR-100. As we can see, the attacks generated by later checkpoints (> 50 epochs) of PGD-AT Baseline lack transferability to the unseen (test) model. This is an insightful result as attacks generated by later checkpoints (corresponding to more robust source models) are supposed to have better transferability (namely, generalization over unseen models). The violation of the above intuition suggests that the defender (outer minimizer in PGD-AT Baseline) starts to overfit the attacker (inner maximizer in PGD-AT Baseline) at later epochs and in turn, is unable to generate attacks with generalization-ability. In AT, the two-layer game makes the defender and the attacker co-evolved and becomes overfitting to each other at later epochs. Indeed, as we use the proposed Baseline + KD + SWA in  Figure 4 (Left), the overfitting issue, characterized by the degradation of attack transferability at later epochs, can largely be mitigated.
>
> [1] Metric Learning for Adversarial Robustness
>
> [2] Rethinking Softmax Cross-Entropy Loss for Adversarial Robustness
>
> [3] Boosting Adversarial Training with Hypersphere Embedding
>
> [4] Revisiting loss landscape for adversarial robustness
>
> [5] Overfitting in adversarially robust deep learning
>
> [6] Decoupling Direction and Norm for Efficient Gradient-Based L2 Adversarial Attacks and Defenses

---

> > ### Comment · AnonReviewer3 · 2020-11-18
> > **Reply to authors' first part of responses**
> >
> > My response to authors’ response [Cons 1: Marginal Contribution.]
> > 1 “Our work for the first time provides principal and effective solutions to mitigate robust overfitting [5] beyond the early stopping.”
> > Recently, I came across some papers. Adversarial training may not stick to minimax formulation. Some curriculum-inspired adversarial training methods may not have the issue of robust overfitting (see [7] [8]).
> > [7] Attacks which do not kill training make adversarial learning stronger.
> > [8] Improving adversarial robustness through progressive hardening.
> >
> > 2  I indeed appreciate the technical contributions on improving the adversarial training.
> > That I state “marginal contribution” comes from the philosophy perspective.
> >
> > My response to authors’ response [Cons 2: Hypothesize Explanation.]
> > 1.I am trying to understand what happened & why robust overfitting occur.
> > In terms of relieving robust overfitting, I also came across the concurrent work of ICLR 2021 submission <https://openreview.net/forum?id=iAX0l6Cz8ub>.
> > Since both are concurrently claim to relieve the robust overfitting, your claiming “solving the problem of robust overfitting” can be deemed as novel.
> >
> > I appreciate your explanations of Figure 4.
> > You are describing the phenomena that “the defender starts to overfit the attacker at later epochs and in turn, is unable to generate attacks with generalization-ability, i.e., co-evolution.”
> > But I am trying to understand what exact things happened behind (e.g., minimax-based AT), which causes these phenomena.
> >
> >
> > I am willing to increase the score if you could provide the convincing arguments.
> > I will carefully look at your second part of responses later.

---

> > > ### Author Response · Authors · 2020-11-20
> > > **Response to Reviewer #3 [More About Cons 1-2]**
> > >
> > > [More about Cons 1: Marginal Contribution] Thanks for the comments and additional references. We agree that robust training does not stick to the min-max formulation, e.g., random smoothing, friendly adversarial training, and adversarial data free robust training (e.g., [15]), though minmax-based adversarial training (AT) remains trendy. However, it is still very interesting and promising to investigate the minmax-based AT since minmax-based AT normally achieves the strongest robustness.
> > >
> > > It is fair to note that our focus is to address the challenge of robust overfitting defined by Rice et al. [5] in minmax-based AT, which also presents state-of-the-art robust performance (superior to [7,8] under the same settings) with the early stopping technique to avoid the curse of overfitting. Although [7,8], as different strategies to solve min-max formulations, are possibly effective in alleviating the robust overfitting, they achieve inferior robustness compared to Rice et al. [5]. This is why we focus on the specific problem in Rice et al. [5] since our findings join (Rice et al. [5]) in re-establishing the competitiveness and superiority of the most commonly-used AT baseline.
> > >
> > > [More about Cons 2: Hypothesis Explanation] As shown in [5], robust overfitting mainly occurs around the epochs of the first learning rate adjustment. Therefore, we conjecture this phenomenon is related to the “two stages hypothesis” in minimization problems [11-14]. These advanced works [11-14] point out that there are two stages in network training. Specifically, from the first to the second stage, the learning rate is decayed to a small magnitude and loss landscapes become much more complicated (e.g., from nearly convex to highly no-convex loss surfaces). However, this hypothesis is hardly explored under the min-max context.
> > >
> > > In the recent work [9], they find that better local linearity (i.e., Equation 5 in [9]) mitigate the catastrophic forgetting in the fast adversarial training. Inspired by [9]’s finding, we also measure the local linearity under the robust overfitting context, as shown in Figure A11. We observe that, during the first learning rate adjustment, the local linearity suddenly decreases. It potentially switches to the second stage of training with a much more complex loss landscape. Then, the properties of adversarial samples may be dramatically changed, due to the different gradients of a harder inner maximization problem. Our proposed methods prevent the sudden drop of local linearity measurements and alleviate the robust overfitting.
> > >
> > > Thank you for updating the score, and your feedback really helped us on improving our work. We will further polish our paper according to your suggestions in the final version.
> > >
> > > [5] Overfitting in adversarially robust deep learning
> > >
> > > [7] Attacks which do not kill training make adversarial learning stronger.
> > >
> > > [8] Improving adversarial robustness through progressive hardening.
> > >
> > > [9] Understanding and Improving Fast Adversarial Training
> > >
> > > [10] Overfitting or Underfitting? Understand Robustness Drop-in Adversarial Training
> > >
> > > [11] The Two Regimes of Deep Network Training
> > >
> > > [12] Towards Explaining the Regularization Effect of Initial Large Learning Rate in Training Neural Networks
> > >
> > > [13] Time Matters in Regularizing Deep Networks: Weight Decay and Data Augmentation Affect Early Learning Dynamics, Matter Little Near Convergence
> > >
> > > [14] Deep Neural Networks are Lazy: On the Inductive Bias of Deep Learning
> > >
> > > [15] Towards Achieving Adversarial Robustness by Enforcing Feature Consistency Across Bit Planes

---

> > > > ### Comment · AnonReviewer3 · 2020-11-23
> > > > **The current explanations are good**
> > > >
> > > > I am satisfied with the added explanations and experiments.
> > > > I believe the authors' responses have further enhanced their work.
> > > > Therefore, I vote for acceptance and increase my score further to 7.

---

> ### Author Response · Authors · 2020-11-18
> **(Continued) Response to Reviewer #3 [Cons 3]**
>
> [Cons 3: More robustness evaluations and more network structures.] Thanks for the suggestion. We add the plots of SWA SA/RA in Figure 3 in the updated draft. SWA alone also helps mitigate robust overfitting but does not outperform KD&SWA. This indicates that both KD and SWA provide supplementary benefits to mitigate robust overfitting. Besides, we are evaluating our approach under the Auto-Attack and CW attack on ResNet-18 (in Table S2), and conducting more experiments with VGG16, Wide-ResNet-34-4, and Wide-ResNet-34-10 (in Table S1). Specifically, as shown in Table S2, our methods are effective under both Auto-Attack and CW Attack. Take the experiment on CIFAR-10 with $\ell_\infty$ adversary as an example, our approach shrinks the robust accuracy gap between the best checkpoint and the final model from 7.04% to -0.09% under Auto-Attack and 14.96% to 0.79% under CW Attack. And Table S1 shows our methods can be generalized across different architectures. All results have been added to this response and our draft.
>
> **Table S1.** Performance showing the occurrence of robust overfitting across different architectures and the effectiveness of our proposed remedies under $\ell_\infty$ PGD-20 adversary. We pick the checkpoint which has the best robust accuracy on the validation set.
>
> |Architecture|Dataset|Settings|Robust Accuray(RA)(Best->Final)| Standard Accuracy(SA)(Best->Final)|
> |:-:|:-:|:-:|:-:|:-:|
> |VGG-16|CIFAR-10|Baseline|(46.42→40.59)(↓5.83)|(75.29→79.54)(↑4.25)|
> |VGG-16|CIFAR-10|Our Methods|(48.99→48.93)(↓0.06)|(79.00→79.69)(↑0.69)|
> |VGG-16|CIFAR-100|Baseline|(21.64→17.43)(↓4.21)|(39.26→45.84)(↑6.58)|
> |VGG-16|CIFAR-100|Our Methods|(24.79→24.73)(↓0.06)|(48.20→49.00)(↑0.80)|
> |WideResNet-34-4|CIFAR-10|Baseline|(52.59→43.06)(↓9.53)|(81.53→83.28)(↑1.75)|
> |WideResNet-34-4|CIFAR-10|Our Methods|(54.28→53.90)(↓0.38)|(85.17→85.50)(↑0.33)|
> |WideResNet-34-4|CIFAR-100|Baseline|(28.02→20.61)(↓7.41)|(53.19→53.63)(↑0.44)|
> |WideResNet-34-4|CIFAR-100|Our Methods|(30.10→29.80)(↓0.30)|(57.23→58.05)(↑0.82)|
> |WideResNet-34-10|CIFAR-10|Baseline|(54.27→47.12)(↓7.15)|(84.16→85.72)(↑1.56)|
> |WideResNet-34-10|CIFAR-10|Our Methods|(55.50→55.34)(↓0.16)|(86.81→87.06)(↑0.25)|
> |WideResNet-34-10|CIFAR-100|Baseline|(29.95→24.02)(↓5.93)|(56.56→56.42)(↓0.14)|
> |WideResNet-34-10|CIFAR-100|Our Methods|(31.93→31.51)(↓0.42)|(60.86→61.78)(↑0.92)|
>
>
> **Table S2.** Robust accuracy under Auto-Attack and CW Attack on CIFAR-10/100 with ResNet18. The best checkpoint is picked with the best robust accuracy under PGD-20 on the validation set. We follow the same setting as [6] for CW Attack: 1 search step on C with an initial constant of 0.1, with 100 iterations for each search step and learning rate is 0.01.
>
> |Dataset|Norm|Settings|Auto-Attack(Best->Final)|CW Attack(Best->Final)|
> |:-:|:-:|:-:|:-:|:-:|
> |CIFAR-10|$\ell_\infty$|Baseline|(47.00→39.96)(↓7.04)|(75.48→60.52)(↓14.96)|
> |CIFAR-10|$\ell_\infty$|Our Methods|(49.35→49.44)(↑0.09)|(77.83→77.04)(↓0.79)|
> |CIFAR-10|$\ell_2$|Baseline|(67.18→64.29)(↓2.89)|(73.80→53.77)(↓20.03)|
> |CIFAR-10|$\ell_2$|Our Methods|(68.87→68.90)(↑0.03)|(73.89→73.79)(↓0.10)|
> |CIFAR-100|$\ell_\infty$|Baseline|(22.73→18.11)(↓4.62)|(45.89→37.76)(↓8.13)|
> |CIFAR-100|$\ell_\infty$|Our Methods|(25.42→25.35)(↓0.07)|(49.46→49.07)(↓0.39)|
> |CIFAR-100|$\ell_2$|Baseline|(37.16→33.43)(↓3.73)|(48.43→37.73)(↓10.70)|
> |CIFAR-100|$\ell_2$|Our Methods|(40.56→40.61)(↑0.05)|(51.02→50.90)(↓0.12)|
>
> [6] Decoupling Direction and Norm for Efficient Gradient-Based L2 Adversarial Attacks and Defenses

---

> > ### Comment · AnonReviewer3 · 2020-11-19
> > **The added experiments look promising.**
> >
> > The added experiments across networks structures & attack evaluations look good. \
> > Therefore, I am willing to increase the score to 6.

---

### Official Review · AnonReviewer1 · 2020-10-28
**Studies smoothing approaches from standard training to reduce robust overfitting**

**Rating:** 7
**Confidence:** 4

**Review:**

Summary
=======
The paper leverages two methods for improving generalization in standard training, logit smoothing and stochastic weight averaging, and show that these results can mitigate robust overfitting and improve generalization for adversarial training methods.

Overall, the paper was clear and easy to follow. There are a number of ablation studies showing the marginal effects of the two methods, as well as experiments demonstrating how the approaches vary with certain choices in methodology. My initial impression is positive, though there are certain changes described below which would help solidify the paper and its claims.


Comments for discussion
=======================
By improving upon the results in Rice et al. 2020, the authors purport to have state of the art results. However, there's be a plethora of new work since then which have improved these numbers even further. Fortunately, since the submitted work handles the standard CIFAR10 setting, there are a number of public benchmarks that can be used here. It would be great if the authors could train a comparable model and evaluate it using one of these benchmarks (e.g. the autoattack framework at https://github.com/fra31/auto-attack).

To be clear, since a number of these approaches on the benchmark are quite recent, I am not requesting that the authors directly compare to these new methods in their work. However, the baseline that they do compare to (e.g. Rice et al. 2020) is evaluated in this framework (and had a not-insignifcant drop in robust accuracy), so it would be of significant utility to also evaluate the approach using the improved attack. Performing this evaluation would serve two purposes:

1. This should alleviate most concerns on the validity of the result
2. This makes the work easily comparable for future work

Note that reaching the top of the benchmark is not a requirement for publication. As long as it is consistent with the claims of the paper, that the approach reduces robust overfitting for PGD training and improves upon the PGD baseline within this benchmark, then this is fine. If the authors can report how their approach performs under this improved evaluation or a comparable alternative, then I am happy to adjust my score accordingly. However, the authors probably shouldn't claim state-of-the-art performance without doing this evaluation first.


Minor comments
==============
+ In section 3.2, it is mentioned that Table 2 supposedly shows differences when a robust self-teacher is pretrained, but this does not seem to be the case.


Update
======
I have looked through the response and edited version. The updated evaluation looks solid and provides a potential solution to a robust overfitting problem. Although the work is primarily empirical in nature, it may inspire directions for future work to look into more theoretical explanations of robust overfitting.

---

> ### Author Response · Authors · 2020-11-20
> **Response to Reviewer #1 [Cons 1-2]**
>
> We’re very glad you had a positive initial impression, and likewise, we found the set of perceptive questions you raised in your feedback very insightful, pushing us to think of a tighter experiment design. We provide pointwise responses below.
>
> [Cons1: Evaluation under the Improved Attack] Thanks for the helpful suggestion.  We understood your concerns on the evaluation of our approach under improved attacks, e.g., Auto-Attack and CW attack. We have included the reference and experiment results of the Auto-Attack and CW attack in our modified draft. The Auto-attack and CW attack implementations that we used follow (https://github.com/fra31/auto-attack) and the advertorch (https://github.com/BorealisAI/advertorch), respectively. As shown in Table S1, after applying the combination of KD and SWA, the overfitting problem is largely mitigated under Auto-Attack and CW attack. Take CIFAR-10 $\ell_\infty$ adversary as an example.  As we can see, our approach reduces the drop of robust accuracy from 7.04% to -0.09% under Auto-Attack, and 14.96% to 0.79% under CW attack, when comparing the best model to the eventually converged model.  Although the Auto-Attack Leaderboard is a rising benchmark, it is hard to conduct a fair comparison since there are diverse settings, including extra data, network architecture, training epochs, and other implementation configurations. We try our best to unify the setting and compare it with the ResNet-18 models from Rice et al. [1] in the Auto-Attack Leaderboard (https://github.com/fra31/auto-attack). We observe the ResNet-18 models from [1] achieve 18.95% robust accuracy with $\ell_\infty$ adversary auto-attacks on CIFAR-100 and 67.68% robust accuracy with $\ell_2$ adversary on CIFAR-10, while our approaches reach 25.42% and 68.87% robust accuracy, respectively. Under the exact same settings, our proposal achieves substantial robustness improvement (6.47% and 1.19% robust accuracy) under Auto-Attacks. In addition, we have also provided additional experiment results on Wide-ResNet in the modified draft and submit all our models to the Auto-Attack leaderboard in the future.
>
> [Cons2: Typo] Thanks for the careful reading. Table 2 was a typo. It should be Table 4.
>
> [1] Overfitting in adversarially robust deep learning
>
> [2] Decoupling Direction and Norm for Efficient Gradient-Based L2 Adversarial Attacks and Defenses
>
> Table S1 Robust accuracy under Auto-Attack and CW Attack on CIFAR-10/100 with ResNet18. The best checkpoint is picked with the best robust accuracy under PGD-20 on the validation set. We follow the same setting as [2] for CW Attack: 1 search step on C with an initial constant of 0.1, with 100 iterations for each search step and learning rate is 0.01.
>
> |Dataset|Norm|Settings|Auto-Attack(Best->Final)|CW Attack(Best->Final)|
> |:-:|:-:|:-:|:-:|:-:|
> |CIFAR-10|$\ell_\infty$|Baseline|(47.00→39.96)(↓7.04)|(75.48→60.52)(↓14.96)|
> |CIFAR-10|$\ell_\infty$|Our Methods|(49.35→49.44)(↑0.09)|(77.83→77.04)(↓0.79)|
> |CIFAR-10|$\ell_2$|Baseline|(67.18→64.29)(↓2.89)|(73.80→53.77)(↓20.03)|
> |CIFAR-10|$\ell_2$|Our Methods|(68.87→68.90)(↑0.03)|(73.89→73.79)(↓0.10)|
> |CIFAR-100|$\ell_\infty$|Baseline|(22.73→18.11)(↓4.62)|(45.89→37.76)(↓8.13)|
> |CIFAR-100|$\ell_\infty$|Our Methods|(25.42→25.35)(↓0.07)|(49.46→49.07)(↓0.39)|
> |CIFAR-100|$\ell_2$|Baseline|(37.16→33.43)(↓3.73)|(48.43→37.73)(↓10.70)|
> |CIFAR-100|$\ell_2$|Our Methods|(40.56→40.61)(↑0.05)|(51.02→50.90)(↓0.12)|

---

### Author Response · Authors · 2020-11-23
**General Response**

We sincerely appreciate all reviewers’ time and efforts in reviewing our paper. We truly thank reviewer #3 for the acknowledgment of our extra experiments and explanations, and for increasing the score of our paper. We genuinely appreciate the positive initial impression from reviewer #1 and #2. And we also thank all reviewers for their insightful and constructive suggestions, which help a lot in further improving our paper.

In addition to the pointwise responses below, here we summarize our updates.

- **[Extra Experiments]** As mentioned by all three reviewers, we conduct new experiments to evaluate our approaches under improved attacks, e.g. Auto-Attack and CW attack. In the meantime, we apply our methods with VGG-16, Wide-ResNet-34-4, and Wide-ResNet-34-10. The results demonstrate that our approaches are effective under improved attacks and can be generalized to different models.

- **[Modified Draft]** The modified nine pages draft is updated, including new experiment results and references. We will keep updating for better and clear readability.

- **[Reproducibility]** Both training and evaluation codes for KD&SWA have been provided as additional supplementary material and the pre-trained models with ResNet18 on CIFAR10 can be found at https://www.dropbox.com/sh/1htrkwawfqh2hem/AAAOIz2A7ndA9VLyA6X_BXSta?dl=0.

We hope our pointwise responses below could clarify all reviewers’ confusion and alleviate all concerns. We thank all reviewers’ time again.

---

### Decision · Program_Chairs · 2021-01-07
**Final Decision**

**Decision:**

Accept (Poster)

**Comment:**

This paper focuses on the problem of robust overfitting. The philosophy behind sounds quite interesting to me, namely, injecting
more learned smoothening during adversarial training. This philosophy leads to two simple yet effective methods: one leveraging knowledge distillation and self-training to smooth the logits, and the other performing stochastic weight averaging to smooth the weights.

The clarity and novelty are above the bar of ICLR. While the reviewers had some concerns on the significance, the authors did a particularly good job in their rebuttal. Thus, all of us have agreed to accept this paper for publication! Please carefully address all
comments in the final version.